# Spatial and Temporal Differentiation of the Tourism Water Footprint in Mainland China and Its Influencing Factors

**Zhenquan Xiao [1,2], Polat Muhtar [1,2,\*], Wenxiu Huo [3], Chaogao An [1,2], Ling Yang [2] and Fengrong Zhang [2]**

1   Key Laboratory of the Sustainable Development of Xinjiang's Historical and Cultural Tourism, Xinjiang University, Urumqi 830046, China; zqydxiao@sina.com (Z.X.); anchaogaohan@sina.com (C.A.)
2   School of Tourism, Xinjiang University, Urumqi 830049, China; yl_yang_ling@sina.com (L.Y.); frzhangrr@sina.com (F.Z.)
3   Anshan Vocational and Technical College, Anshan Normal University, Anshan 114000, China; huowenxiuhappy@sina.com
\*   Correspondence: pulati_mot@xju.edu.cn

**Abstract:** While tourism generates economic benefits at destinations, it also creates certain environmental pressures. In the global context of water scarcity, the spatial and temporal differentiation characteristics of water consumption at tourism destinations have become a focus of attention. Based on panel data, the present study calculates the change trends in China's tourism water footprint (*TWF*) in the 2013–2018 period using input-output analysis, analyses the regional differences in *TWF* changes using kernel density estimation and the Theil index, and investigates the driving factors of the spatial and temporal differentiation of the *TWF* using the logarithmic mean Divisia index model. The results indicate that (1) the tourism water consumption in China increased year-by-year but that the tourism water use efficiency improved; (2) the proportion of the *TWF* for accommodation and food in the total *TWF* gradually increased, while the proportion of the *TWF* for transportation continuously decreased; (3) the *TWF* of each region increased continuously, with the absolute difference between regions gradually increasing and the difference in the *TWF* intensity gradually decreasing; and (4) decomposition analysis showed that the *TWF* in China was positively driven by per capita expenditure and the number of tourists, with the role of *TWF* intensity shifting from inhibition to promotion, and that each driving force changed with time. Based on the spatial and temporal differences in the *TWF*, the provinces in China are divided into five categories, and targeted countermeasures and suggestions are proposed.

**Keywords:** tourism water footprint; the spatial and temporal differentiation; input-output analysis; logarithmic mean Divisia index model; China

## 1. Introduction

Water resources are the material basis for the survival and development of human society, and water security is of great significance for ensuring ecological security and sustainable socio-economic development [1]. Global warming will lead to an increase in rainfall and will accelerate the hydrological cycle, and renewable freshwater resources are expected to grow faster than water demand [2]; therefore, the water supply seems to be sufficient to meet human needs, but this is not the case if other factors are considered. Due to factors such as population growth, economic development, and shifting consumption patterns, the global demand for water resources is growing at a rate of 1% per year, and this rate is expected to accelerate significantly in the next 20 years [3]. Because of the casual discharge of industrial pollutants, seawater intrusion, and increased agricultural activities [4–6], surface water and groundwater are polluted, available water resources are dwindling, and both quality- and quantity-induced water scarcity problems are becoming increasingly serious. It is estimated that by 2030, the Earth will face a 40% shortage [7]. Freshwater scarcity has already posed a serious threat to the economic and social

development of many countries [8]; thus, increasing the sustainability of water resources has become an urgent need.

Freshwater is an important resource for tourism [9]. Tourism demand and the sustainability of any tourism destination depend on the availability of water [10], and water is one of the determining factors in the tourism life cycle mode [11,12]. The direct use of water is required for tourist bathing, garden irrigation, and hotel bed linen cleaning, and water consumption is embedded behind food, roads, products, fuel, and other tourist-related services [13,14]. Recreational activities such as swimming, diving, and fishing take place in lakes or seas and are also an important part of the tourism process, all leading to an increase in tourism wastewater [15,16]. Many forms of tourism rely on water, for example, agro-tourism, winter tourism, and coastal tourism. In 2019, the total number of global tourist arrivals reached 12.31 billion, an increase of 4.6% over the previous year, and the total tourism revenue was 5.8 trillion U.S. dollars, equivalent to 6.7% of global GDP [17]. Although the development of tourism has promoted economic growth to a certain extent, frequent human tourism activities inevitably cause different degrees of damages to the environment of tourist attractions, especially when the local relevant departments lack management experience and only pursue economic benefits [18]. The over-abstraction of groundwater due to the dramatic increase in tourism may trigger a land subsidence crisis, thus affecting the sustainability of tourist destinations [19]. Tourists will shift their water demand to other areas, especially some water-scarce areas [20]. In many countries, tourists can even consume more water than local residents [21], reaching 2–3 times the amount of destination residents [22], further exacerbating the existing water scarcity problem [20].

The effective use of water resources has become an enormous challenge for sustainable tourism development [23]. Therefore, many researchers have started to focus on tourism and water relations mainly in the following four areas. The first research area is virtual water and virtual water trade [24,25]. Virtual water refers to the amount of water resources used in the production of goods or services [26]. Virtual water trade is formed when water-intensive products are transferred regionally through trade [25]. The study of virtual water trade between different regions can link regional water resource consumption to water resource allocation and help in understanding the global properties of freshwater resources and the impact of inter-regional consumption and trade on water resource use [27]. The second research area is the tourism water footprint (*TWF*) [13,28,29]. Unlike studies of virtual water that investigate water use from a production perspective, the water footprint (WF) extends the research perspective to consumers. It adds the calculation of the grey WF to the calculation of the blue WF and green WF of virtual water and is a better water use indicator [30]. The grey WF refers to the volume of water required to dilute the contaminants that accompany the production process [31]. The blue WF is defined as the surface and groundwater that is consumed (water that has evaporated and the water that was incorporated into the product) along the value chain of a product; and all the rainwater that does not become run-off, but is consumed, represents the green water footprint [32]. It links virtual water and human consumption and opens up a new field of virtual water research. The third research area is the water-energy-food nexus in tourism [29,33]. Tourists consume more food, energy, and water at their destinations than at their residences. Water, food, and energy serve as the main resources to support sustainable economic, social, and environmental development. With the rapid development of tourism, the demand for water-energy-food resources continues to increase. Most previous studies have investigated single independent elements or established dual local linkages between two elements among water, food, and energy [34,35]; however, few studies have focused on the relationship among the three elements in tourism. The fourth research area is the tourism water resource carrying capacity [36,37]. Buckley [38] argued that the tourism environmental carrying capacity is primarily an ecological concept that refers to the maximum amount of tourism that an ecosystem at a tourism destination can sustain before an irreversible ecological change occurs. Calculating the tourism water

resource carrying capacity can provide a theoretical basis for limiting the intensity of tourism development in scenic areas or destinations.

Hoekstra [39] introduced the concept of the WF, which is the amount of water resources required to produce the goods and services consumed by a certain population under a certain material standard of living. Based on virtual water [26], the WF can track the direct and indirect freshwater demand in production and service processes [25] to help in further understanding the relationship between socio-economic development and the water environment [40]. Tourism belongs to the tertiary industry and is a service industry that is closely related to water resources. The *TWF* is a specific application of the WF theory in the field of tourism that is used to evaluate the real water consumption in tourism. Initially, researchers mainly investigated the accommodation sector [41–43]. With continuous improvements in statistical data, subsequent studies began to cover sectors such as transportation and sightseeing. In terms of research scales, the *TWF* has been calculated at scenic [44,45], urban [28], national [46], and global [20] scales. Based on water source types and pollution types, the *TWF* can be divided into blue, green, and grey WFs [47]. The *TWF* can be divided into direct and indirect WFs based on the correlation between the goods production process and water consumption [9,48,49]. Consumers are mainly concerned about direct water use for showering, garden irrigation, and toilet flushing without understanding water consumption in the goods production process [50]. Initially, researchers also focused on direct water use, such as water use in hotels and for swimming pools and golf courses [51–53]. Gössling et al. [54] noted that indirect water use for tourism might be more important than direct water use; however, there is insufficient research on indirect water use. Some studies have also shown that the indirect WF is much larger than the direct WF [29,55]. Therefore, the indirect WF or the total WF appears to be more reflective of water use in tourism, and water management policies based on these indicators can be more effective.

The WF is a life cycle analysis (LCA) indicator that is used to analyze the water resource demand of a product or service throughout its entire life cycle [56]. Currently, there are two main approaches to calculate the WF: the bottom-up approach and the top-down approach. The bottom-up approach is based on the life cycle theory and directly starts from tourism activities to calculate the resource consumption and waste discharge of tourism products or services [57]. It provides a basis for indirect environmental burden assessments and is considered the most comprehensive method for assessing and comparing materials, products, and services from an environmental perspective [58]. However, LCA requires data acquisition through destination surveys, which are difficult to carry out. Therefore, LCA is usually used to conduct sectoral or small regional TF evaluations [59–61]. In addition, the use of the bottom-up approach to studying the entire tourism industry may lead to poor accuracy of the results due to the difficulty of obtaining all the required information [62,63]. The top-down approach includes the input-output (IO) model and the general equilibrium model [64]. The IO method is based on data mainly derived from official statistics on, for example, energy consumption, water price, and tourist consumption, which are easily accessible and, hence, preferred for estimating the TF at regional, national, and global scales [65–67]. Therefore, to reflect the water consumption of all sectors of the tourism industry, the present study uses a top-down approach in combination with an IO table to investigate the changes in the *TWF* in China.

China consistently ranks second among T20 countries in total tourism revenue [17], with tourism contributing more than 10% to its GDP. From 2011 to 2019, China's domestic tourism revenue and international tourism foreign exchange revenue increased by 200% and 171%, respectively, and the rapid development of tourism has led to rising demand for water in tourism. Additionally, China is a country with severe droughts and water shortages, with a per capita water resource only a quarter of the world average, making it one of the poorest countries in the world in terms of per capita water resources. Water scarcity has always been an important social issue in China and one of the factors limiting the development of Chinese society [68]. The contradiction between the supply

and demand of water resources in China continues to intensify, and water resources are under tremendous pressure. Alleviating the pressure on water resources, addressing the contradiction between supply and demand, and efficiently using water resources are key challenges for sustainable tourism development. In view of this, the Chinese government has promulgated a series of policy documents, proposing to prioritize water conservation and water resource utilization efficiency improvements and emphasizing the need for "water conservation, spatial balance, systematic management, and coordination of the roles of the visible hand (government) and the invisible hand (market)" in the water management process. Academics have also conducted studies on tourism water use in China [47,69,70].

Overall, although the research on the *TWF* is becoming increasingly in-depth, there are still several deficiencies. In terms of the study area, China, as a world tourism power and a country with extreme water scarcity, experiences much more impact by tourism on its domestic and global water environment than does other regions, but the *TWF* in China has rarely been studied. In terms of research perspectives, most studies have focused on static analyses at the global or national level. Although water is a global issue, water shortages are local in nature [48], and differences in development levels and social values in different regions lead to different local water demands [71]. In tourism, the water pressure brought by tourism activities also tends to be spatial and temporal in nature [11]. A static analysis at the global or national level may conceal the spatial and temporal differences in tourism water demand across regions, thereby making it necessary to investigate the variability in the *TWF* across provinces and regions in China from a spatial and temporal perspective. In terms of impact factors, numerous studies have focused on tourism carbon emissions and influencing factors [72–74], only a few studies have specifically addressed the driving forces of the *TWF*. Therefore, this research intends to study the tourism water footprint of 31 provinces in China from 2013 to 2018, improving the tourism water footprint research in the field and perspective. In the calculation process of *TWF*, the theory of six elements of tourism and input and output are combined, and the water expenditure is converted into the volume of water consumption by using the water price of each sector, which enriches the calculation methods and theories of *TWF*. The present study uses IO analysis, kernel density estimation, the Theil index, and the logarithmic mean Divisia index (LMDI) method, combined with provincial IO tables and relevant statistical data, to investigate the spatial differentiation and influencing factors of the *TWF* for China's tourism industry in the 2013–2018 period, with the goal of providing a reference and guidance for alleviating the pressure on regional tourism water resources, scientifically planning and managing regional tourism, and promoting regional coordination. The main objectives of the present study are (1) to use IO analysis to calculate the *TWF* and its intensity in each province; (2) to analyze the spatial and temporal differences in the changes in China's *TWF* from multiple perspectives and to analyze the sources of these differences; (3) to investigate the impact of various driving forces on the changes in the *TWF* in China and each province; and (4) to propose recommendations for tourism water management to promote sustainable tourism development. The remainder of the article comprises the methods and data, results, and discussion, and conclusion.

## 2. Materials and Methods

### 2.1. Data

#### 2.1.1. Total Consumption Coefficients

The basic data for IO analysis are from the Input-Output Table (IO table), which expresses the industry output as the sum of the interindustry flows (transaction between sectors to sectors) and sales to final demand (transaction from sectors to final consumer) [29]. Through the IO table, the total consumption coefficient and then the *TWF* of each sector are obtained. In the China Regional Input-Output Tables (2012) [75], we use the (i) accommodation and food, (ii) transportation, warehousing and postal services, water conservancy, and environmental and public facility management, (iii) wholesale and retail, and (iv) culture, sports and entertainment sectors and their corresponding water production and supply as

the basic data for determining the WFs of food and accommodation, transportation, sight-seeing, and shopping, respectively. In particular, the water conservancy and environmental and public facility management sector are chosen as the data for determining the WF of sightseeing. This decision was made because the explanation and codes for the product sector and industry sector classifications in the China Input-Output Tables (2012) detail that public facility management includes greening management (which refers to the management activities of urban green spaces, green production spaces, protective green spaces, and attached green spaces), urban park management (which refers to the management activities of various types of urban parks that mainly provide people with leisure, viewing, sightseeing and popular science activities) and tourist attraction management (which refers to the management of scenic areas, forest parks, and other tourist attractions), all of which are tourism activities directly related to sightseeing. The tourism-related sectors identified in the present study are provided in Table 1.

**Table 1.** Sectoral breakdown of the tourism industry based on IO tables.

| Account | Sector (IO Table) |
| --- | --- |
| Food and accommodation | Accommodation and food |
| Transportation | Transportation, warehousing, and postal services |
| Sightseeing | Water conservancy and environmental and public facility management |
| Shopping | Wholesale and retail |
| Entertainment | Culture, sports, and entertainment |

### 2.1.2. Tourism Data

The revenue of each sector is needed in the calculation of the *TWFs* of different sectors, and it can be calculated from the total tourism revenue and the proportion of tourism revenue of each sector. We obtained the tourism revenue and the number of tourists in the 2013–2018 period from the Statistical Bulletin of National Economic and Social Development for each province. Because there are no data on the proportion of sectoral revenue by province, we instead use the per capita daily consumption of inbound tourists by region, obtained from the China Tourism Statistical Yearbook (2014–2018) and the China Culture and Tourism Statistical Yearbook (2019). Among them, long-distance transportation, postal and telecommunication, and intra-city transportation expenditures are combined as transportation expenditures.

### 2.1.3. Water Price Data

We use the price of non-residential water in each provincial capital as the unit price of water in the corresponding province; the price information is derived from the China Water website (https://www.h2o-china.com/price/ (accessed on 1 March 2021)). Notably, tiered water pricing is implemented in cities such as Beijing and Shanghai. To unify the standard for easy accounting, we convert tiered water prices to an average water price.

### 2.2. Methods

#### 2.2.1. IO Model

IO analysis was proposed by Leontief [76] to analyse the relationship between economic inputs and outputs when multiple sectors are involved. If an economy has $n$ economic sectors, the following equation can be written based on the equilibrium relationship in the rows of the IO table:

This is example 1 of an equation:

$$AX + Y = X \tag{1}$$

$$X = (I - A)^{-1}Y \tag{2}$$

where *X*, *Y*, and *A* are the total output matrix, the final demand matrix, and the direct consumption coefficient matrix, respectively. $(I - A)^{-1}$ is the Leontief inverse matrix, and *I* is the identity matrix of the same order as *A*.

The direct and total water consumption coefficients of sector *i* are calculated as follows [29]:

$$\tau_k^d = W_k / X_k \tag{3}$$

$$\tau_k^t = (I - A)^{-1} W_k / X_k \tag{4}$$

where $\tau_k^d$ is the direct water consumption coefficient of sector *k*, i.e., the value of direct water consumption per unit of output; $W_k$ is the direct water consumption of sector *k*; $X_i$ is the total output of sector *k*; and $\tau_k^t$ is the total water consumption coefficient of sector *k*, i.e., the total water consumption per unit of output.

The total WF of sector *k*, $wf_k$ is calculated as follows:

$$WF_k = \tau_k^t R_k / P_k \tag{5}$$

where $R_k$ is the revenue of sector *k*, and $P_k$ is the price of water used by sector *k*.

In the present study, the six elements of tourism—food, accommodation, transportation, sightseeing, shopping, and entertainment—are divided into five main accounts, namely, the WF of food and accommodation, the WF of transportation, the WF of sightseeing, the WF of shopping and the WF of entertainment, which are then added to obtain the total *TWF*, with the specific equations as follows:

$$TWF_k = \tau_k^t Rr_k / P_k \tag{6}$$

$$TWF = \sum_{i=1}^{5} TWF_k \tag{7}$$

where $TWF_k$ is the *TWF* of account *k*, *R* is the total tourism revenue of the region, $r_k$ is the percentage of tourism revenue of account *k* in the total tourism revenue, $T_k$ is the total water consumption coefficient of account *k*, $P_k$ is the unit price of water for the service sector operating in the region.

### 2.2.2. Analysis of WF Intensity

*TWF* intensity is defined as the ratio of the *TWF* to tourism revenue. It can be used to measure the water use efficiency of tourism, i.e., the water use efficiency is low when the *TWF* intensity is high. It is calculated as follows:

$$S_j = TWF_j / R_j \tag{8}$$

where *j* represents a province, $S_j$ is the WF intensity, and $R_j$ is the annual tourism revenue of province *j*.

### 2.2.3. Kernel Density Estimation

Kernel density estimation is a nonparametric method for estimating probability density functions and has the advantages of arbitrary functional forms and few restrictions on variable distributions [77]. To investigate the distribution characteristics and dynamic evolution of the regional *TWF* in China, kernel density estimation is used in the present study to investigate the evolution of the *TWF* in each region. Assuming that $z_1$, $z_2$, $z_3$, and ... $z_n$ have an identical distribution, the kernel density function can be expressed as:

$$F(z) = \frac{1}{h} \sum_{i=1}^{n} K\left(\frac{z_i - \bar{z}}{h}\right) \tag{9}$$

where $K(\cdot)$ is the kernel function, $z_i$ is an observation, and $\bar{z}$ is the mean of observations.

### 2.2.4. Theil Index

The Theil index is a method used to measure regional disparities and was originally adopted in the study of regional differences in income levels. The larger the value of the Theil index, the greater is the regional difference. In the present study, the Theil index is introduced to study the *TWF*, and the Theil index for regional differences in the WF and WF intensity of tourism in China is constructed, with its calculation and decomposition expressed in the following equations [78]:

$$T_{inter} = \sum_{i=1}^{n} \left( \frac{TWF_i}{TWF} \right) \ln \left( \frac{TWF_i / TWF}{R_i / R} \right) \tag{10}$$

$$T_{intra} = \sum_{i=1}^{n} \left( \frac{TWF_i}{TWF} \right) \sum_{j=1}^{n} \left( \frac{TWF_{ij}}{TWF_i} \right) \ln \left( \frac{TWF_{ij} / TWF_i}{R_{ij} / R_i} \right) \tag{11}$$

$$T = T_{inter} + T_{intra} \tag{12}$$

where $TWF_i$ is the *TWF* of region $i$; $TWF_{ij}$ is the *TWF* of province $j$ in region $i$; $R$ represents tourism revenue; $T$, $T_{inter}$ and $T_{intra}$ represent the overall, inter-regional, and intra-regional differences, respectively.

In addition, the Theil index can also be used to analyze the causes of the overall regional differences by calculating the inter-regional contribution ($C_{inter}$) and the intra-regional contribution ($C_{intra}$), which are calculated using the following equations:

$$C_{inter} = T_b / T \tag{13}$$

$$C_{intra} = T_w / T \tag{14}$$

### 2.2.5. LMDI Model

Decomposition analyses are techniques that are used to evaluate historical changes in economic, environmental, and other socio-economic indicators to trace the underlying factors that contribute to such changes [79]. At present, the most commonly used decomposition analysis methods are structural decomposition analysis (SDA) and index decomposition analysis (IDA). SDA is capable of more refined decompositions of economic and technological effects, but IDA is capable of more detailed time and country studies because of the availability of data [39]. The aim of the present study is to examine the changes in the driving forces each year in China and the differences in the driving forces in different provinces and to investigate the factors influencing the WF growth in a time series. IDA was ultimately chosen after a comprehensive comparison.

Among the multiple decomposition forms of the IDA, the additive form of the LMDI is broadly used due to its simple operation, ease of understanding, and residual-free decomposition [79–81]. Therefore, the present study uses LMDI decomposition to analyze the impact of the intensity effect ($S$), economic effect ($E$), and demographic effect ($Q$) on the changes in the *TWF* in China and in 31 provinces in the 2013–2018 period. The intensity effect measures the contribution of the improvements in the WF per unit of output to the *TWF*; the economic effect measures the impact of changes in the per capita tourist expenditure on the *TWF*; the demographic effect measures the impact of changes in the number of tourists on the *TWF* and can also be considered the final demand effect. The decomposition expression is defined as follows:

$$TWF_n = \frac{TWF_n}{TR_n} \times \frac{TR_n}{TP_n} \times TP_n = S \times E \times Q \tag{15}$$

where $n$ represents a year between 2013 and 2018; $TWF_n$ is the *TWF* in the nth year; $TR_n$ is the total tourism revenue in the $n$th year; $TP_n$ is the number of tourists in the nth year; $S$ represents the *TWF* intensity; $E$ represents the per capita tourist expenditure; $Q$ represents the number of tourists.

Therefore, based on the LMDI method, the changes in *TWF* from year *t* (base year) to year *t* + 1, $\Delta TWF_{total}$, can be expressed as:

$$\Delta TWF_{total} = TWF_{t+1} - TWF_t = \Delta TWF_S + \Delta TWF_E + \Delta TWF_Q \tag{16}$$

where $TWF_{t+1}$ is the *TWF* in year *t* + 1, $TWF_t$ is the *TWF* in year *t*, $\Delta TWF_S$, $\Delta TWF_E$ and $\Delta TWF_Q$ are the impacts of WF intensity, per capita expenditure, and number of tourists on the changes in *TWF*, respectively.

The individual effects on the right-hand side of Equation (16) are calculated as follows:

$$\Delta TWF_S = \omega \ \ln \frac{S^{t+1}}{S^t} \tag{17}$$

$$\Delta TWF_E = \omega \ \ln \frac{E^{t+1}}{E^t} \tag{18}$$

$$\Delta TWF_Q = \omega \ \ln \frac{Q^{t+1}}{Q^t} \tag{19}$$

$$\omega = \frac{TWF^{t+1} - TWF^t}{\ln TWF^{t+1} - \ln TWF^t} \tag{20}$$

A flowchart related to data and methodology is presented in Figure 1 to give a clear understanding of the research ideas.

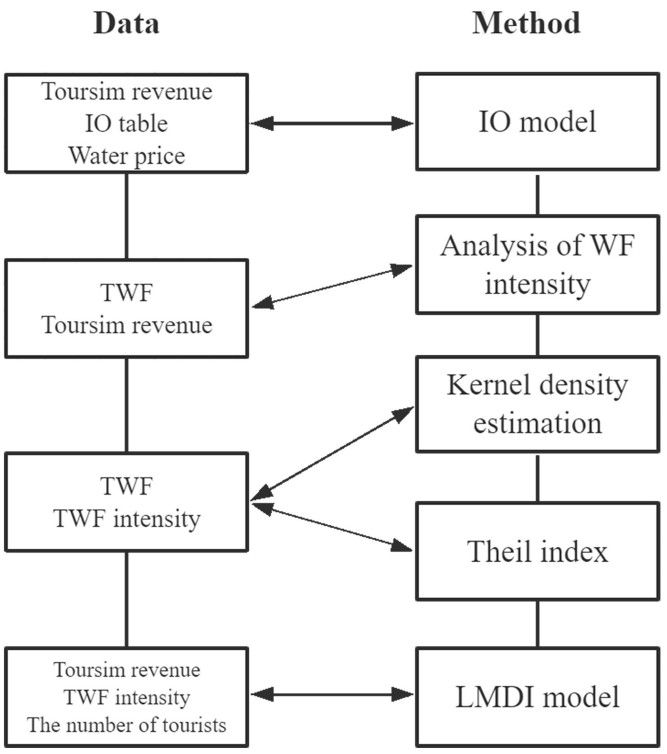

**Figure 1.** Method flow chart.

## 3. Results

### 3.1. Trend Analysis of TWF Changes in China

The *TWF*s of 31 provincial regions in mainland China from 2013 to 2018 are calculated using the IO model. For the sake of comparison, the 31 provincial regions are divided into the Eastern, Central, Western, and Northeast regions (Figure 2).

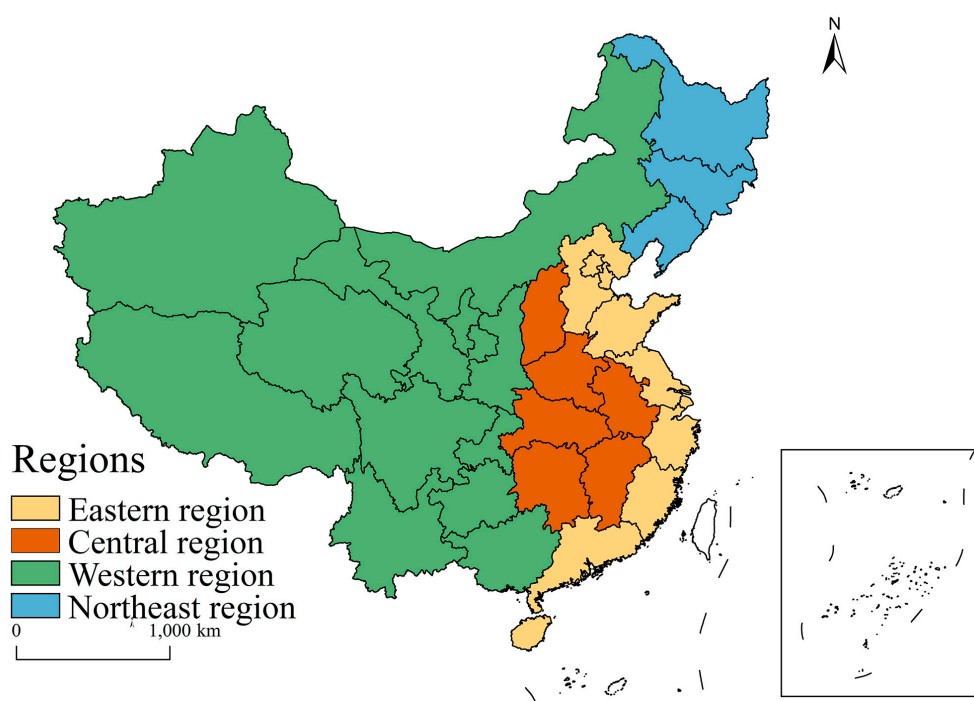

**Figure 2.** The four regions of China. Note: Because of geographical and political reasons, Mainland China is divided into four regions: Eastern, Central, Western, and Northeastern. The Eastern region contains: Beijing, Tianjing, Hebei, Shanghai, Jiangsu, Fujian, Shandong, Zhejiang, Guangdong, Hainan. The Central region contains: Anhui, Shanxi, Jiangxi, Henan, Hubei, and Hunan. The Western region contains: Inner Mongolia, Shannxi, Sichuan, Chongqing, Yunnan, Guizhou, Guangxi, Gansu, Xinjiang, Qinghai, Xizang, and Ningxia. The Northeast region contains: Liaoning, Heilongjiang, and Jilin.

### 3.1.1. Changes in Total *TWF*

The *TWF* in China increased from $120.28 \times 10^8$ m$^3$ in 2013 to $254.87 \times 10^8$ m$^3$ in 2018, an increase of 112%. In addition, the *TWF*s of the Eastern, Central, Western, and Northeast regions each exhibited a fluctuating upward trend, with increases of 112%, 81%, 209%, and 113%, respectively. As seen in Figure 3a, the average WFs of China and different regions showed similar changes during the study period, exhibiting a fluctuating growth trend from 2013 to 2015 and a continuous increase from 2015 to 2018, with an overall upward trend. The *TWF*s differed significantly among regions. Specifically, the *TWF* in the Eastern region was notably higher than that in other regions, and the national average; the *TWF* in the Central region was slightly lower than the national average, and the *TWF*s in the Western and Northeast regions were remarkably lower than the national average.

### 3.1.2. Changes in *TWF* Intensity

Figure 3b shows that during the study period, the *TWF* intensity in China exhibited a fluctuating state and eventually showed a slight downward trend from 0.0015 m$^3$ per yuan in 2013 to 0.0013 m$^3$ per yuan in 2018. The change in the Eastern region was similar to that of the overall sample, and there was a large decrease in the Western and Central regions, indicating an improvement in the water use efficiency of the overall sample and in the three regions. The *TWF* intensity in the Northeast region increased from 0.0007 m$^3$ per yuan to 0.0013 m$^3$ per yuan, with a decrease in tourism water use efficiency.

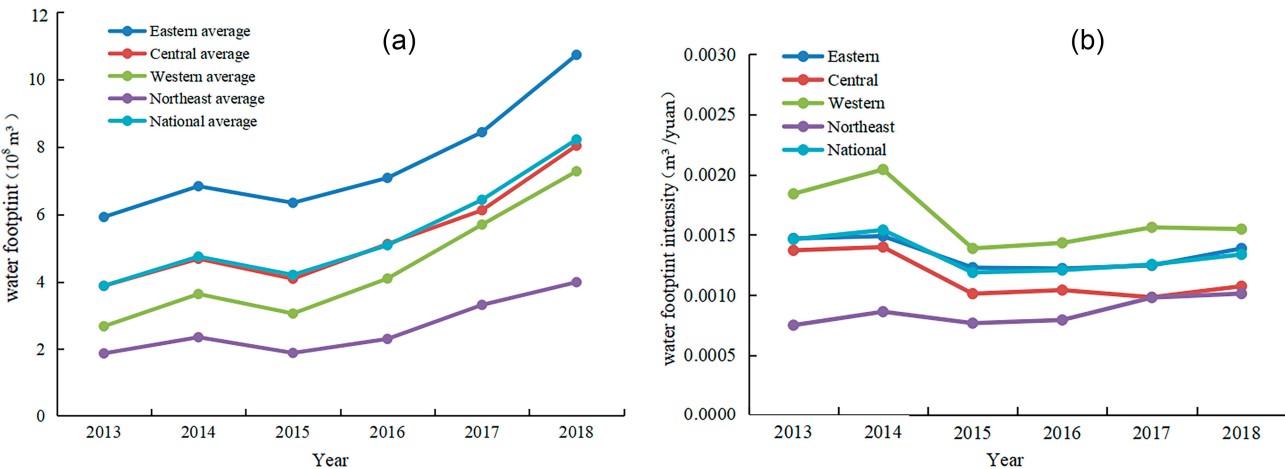

**Figure 3.** (**a**) Overall characteristics of China's *TWF*s in the 2013–2018; (**b**) Overall characteristics of China's *TWF* intensities in the 2013–2018.

### 3.1.3. Changes in *TWF* Composition

Data for 2018 are used as an example to analyze the composition of China's *TWF* (Figure 4). In this year, the total *TWF* in China was $254.9 \times 10^8$ m$^3$; food and accommodation accounted for the largest water use, i.e., 47% ($120.3 \times 10^8$ m$^3$) of the *TWF*, followed by transportation (19%, $48.94 \times 10^8$ m$^3$), entertainment (17%, $42.42 \times 10^8$ m$^3$), shopping (11%, $27.99 \times 10^8$ m$^3$) and sightseeing (6%, $15.99 \times 10^8$ m$^3$), indicating that China's *TWF* mainly originates from food, accommodation, transportation, and entertainment. Basic tourism consumption and non-basic tourism consumption are important indicators of the level of tourism development. Food and beverage, accommodation, transportation, and entertainment constitute basic consumption, while tourism shopping and medical care constitute non-basic consumption. China's tourism industry has developed rapidly and ranked thirteenth in 2019 in overall tourism competitiveness among more than 100 countries, but there is still a disparity in tourism when compared with that of the European Union and other developed countries. During tourism activities, tourists mainly satisfy subsistence consumption needs rather than indulge in luxury consumption.

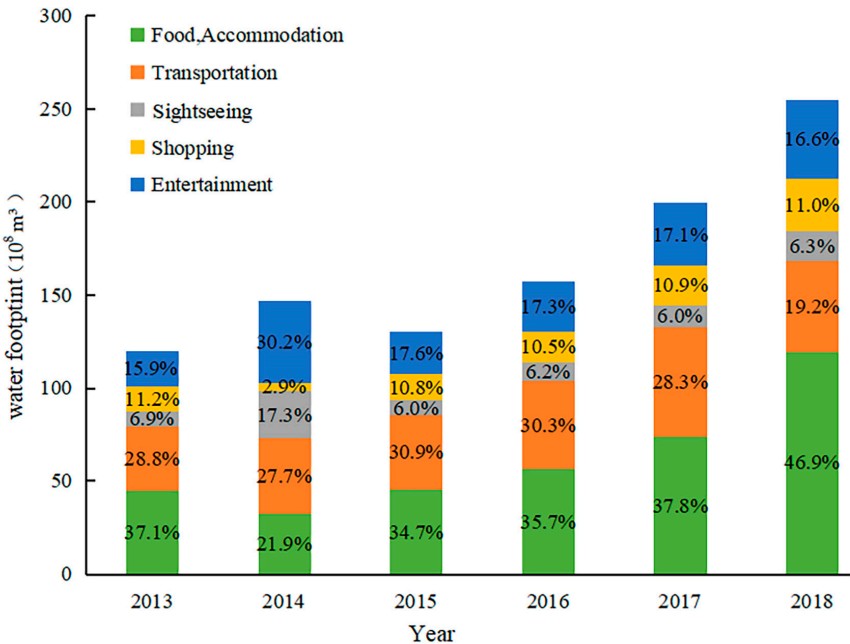

**Figure 4.** China's *TWF* in the 2013–2018 period.

### 3.2. Spatial Difference Analysis of China's TWF

3.2.1. Inter-Provincial Differences in the *TWF*

Total Difference

In the 2013–2018 period, the average increases in the *TWF* of and *TWF* intensity in different provinces were $4.34 \times 10^8$ m$^3$ (144%) and $-0.0002$ m$^3$ per yuan ($-7.01\%$), respectively.

To reflect the spatial evolution of the *TWF* of and *TWF* intensity in each province more intuitively, the present study selects 2013, 2015, and 2018 as time nodes and uses ArcGIS to plot spatiotemporal evolution maps of 31 provinces (Figure 5). The spatial distribution of the *TWF* of and *TWF* intensity in China from 2013 to 2018 showed the following three characteristics.

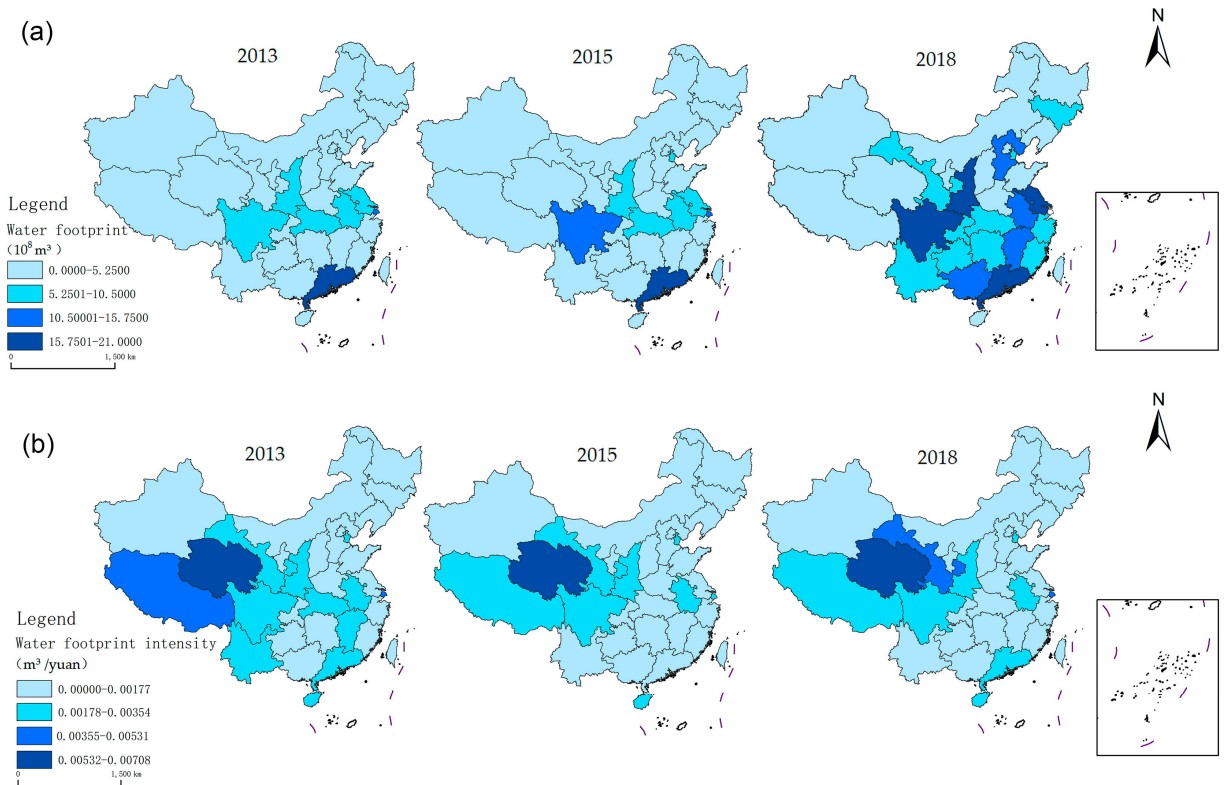

**Figure 5.** (**a**) Spatial distribution of the *TWF* in China; (**b**) Spatial distribution of the *TWF* intensity in China.

First, China's *TWF* exhibited an overall increasing trend, while its *TWF* intensity showed a fluctuating decreasing trend. Specifically, the *TWF* increased in 23 provincial regions from 2013 to 2015 and in all provincial regions from 2015 to 2018, while the *TWF* intensity decreased in 23 provincial regions from 2013 to 2015, in only nine provincial regions from 2015 to 2018.

Second, China's *TWF* exhibited the following spatial distribution pattern: high in provincial regions to the southeast of the Hu line (a straight line connecting Heihe City in Heilongjiang Province and Tengchong City in Yunnan Province) and low in provincial regions to the northwest of the Hu line; the *TWF* intensity showed the opposite pattern.

Third, the provinces with a high *TWF* showed a trend of continuous concentration, while those with a high *TWF* intensity exhibited a trend of slight dispersion. From 2013 to 2018, the *TWF* of provinces such as Jiangsu, Anhui, and Hebei increased continuously, causing most of the provinces with high values to be concentrated in southeastern China. Due to the decrease in the *TWF* intensity in provinces such as Hubei and Jiangxi, the originally concentrated and contiguous areas with high *TWF* intensity values became scattered, but such areas were still concentrated in the Northwest region.

Structural Differences

The spatial distribution of the *TWF* structure in each province is analyzed using the 2018 data as a representative example. Figure 6 shows that in terms of the *TWF*, the top four provinces were Guangdong (28.47 × 10$^8$ m$^3$), Shanghai (23.11 × 10$^8$ m$^3$), Jiangsu (16.21 × 10$^8$ m$^3$), and Sichuan (18.28 × 10$^8$ m$^3$), and the bottom four provinces were Tibet (1.04 × 10$^8$ m$^3$), Xinjiang (0.79 × 10$^8$ m$^3$), Shandong (0.78 × 10$^8$ m$^3$) and Ningxia (0.13 × 10$^8$ m$^3$). In terms of tourism revenue, Guangdong, Sichuan, and Jiangsu were the top provinces, while Xinjiang, Ningxia, and Tibet ranked the lowest, indicating that the *TWF* may be related to the level of tourism economic development. In contrast, Shandong ranked third nationally in tourism revenue but second to last in *TWF*. We speculate that this may be related to the higher water price and low total water consumption coefficient for Shandong. In terms of *TWF* composition, most provinces were consistent with the national *TWF* composition, food and accommodation accounting for the highest proportion and sightseeing the lowest. The WF of entertainment accounted for the highest proportion for Jiangsu, Guangxi, Tibet, and Xinjiang. The results indicate that the *TWF* structure varied among provinces and was also influenced by various factors such as regional transportation conditions, resource endowments, and economic development.

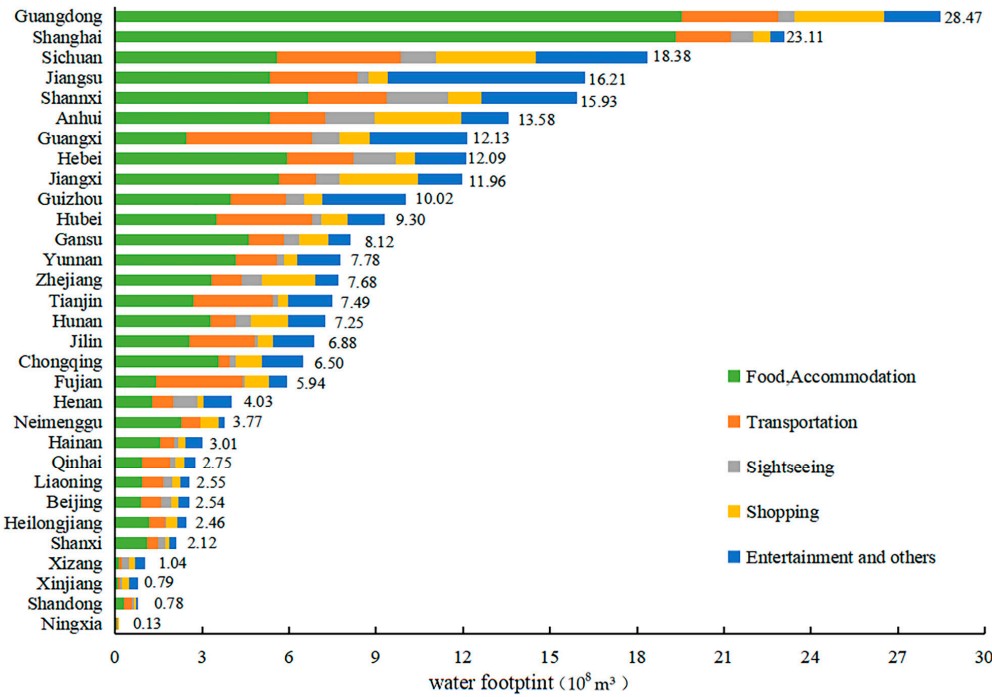

**Figure 6.** *TWF*s of different provincial regions in China in 2018.

3.2.2. Regional Differentiation of the *TWF*

Kernel Density Analysis

Figures 7 and 8 (drawn by Stata) show the dynamic evolution of the kernel density of the *TWF* and *TWF* intensity, respectively, for each region in China from 2013 to 2018.

As shown in Figure 7, the kernel density curves for the *TWF* in the Eastern and Western regions exhibit a clear right-skewed distribution, with the main peaks gradually decreasing and shifting to the right and the width of the main peaks gradually increasing, indicating that the absolute difference in the *TWF*s continued to expand during this period. In particular, the kernel density curve for the Eastern region in 2013 shows a bimodal pattern, but in the curve for 2015, one peak disappears, and the curve becomes flatter, indicating that a polarization pattern for the *TWF* in the Eastern region gradually weakened. The curve for the Western region changes from a bimodal pattern to a trimodal pattern and finally to a monomodal pattern, indicating that the extreme differentiation pattern for

the *TWF* in the Eastern region deepened and then weakened. The kernel density curve for the Central region also shows a shift to the right year-by-year with the width increasing continuously, indicating that the *TWF* increased continuously and the absolute difference between provinces expanded continuously. The kernel density for the Northeast region first shifts slightly to the left and then to the right, the peak decreases, the shape smooths, and the change interval increases significantly, indicating that the absolute difference in WF in the Northeast region during the observation period first decreased and then increased but had an overall trend of gradual expansion.

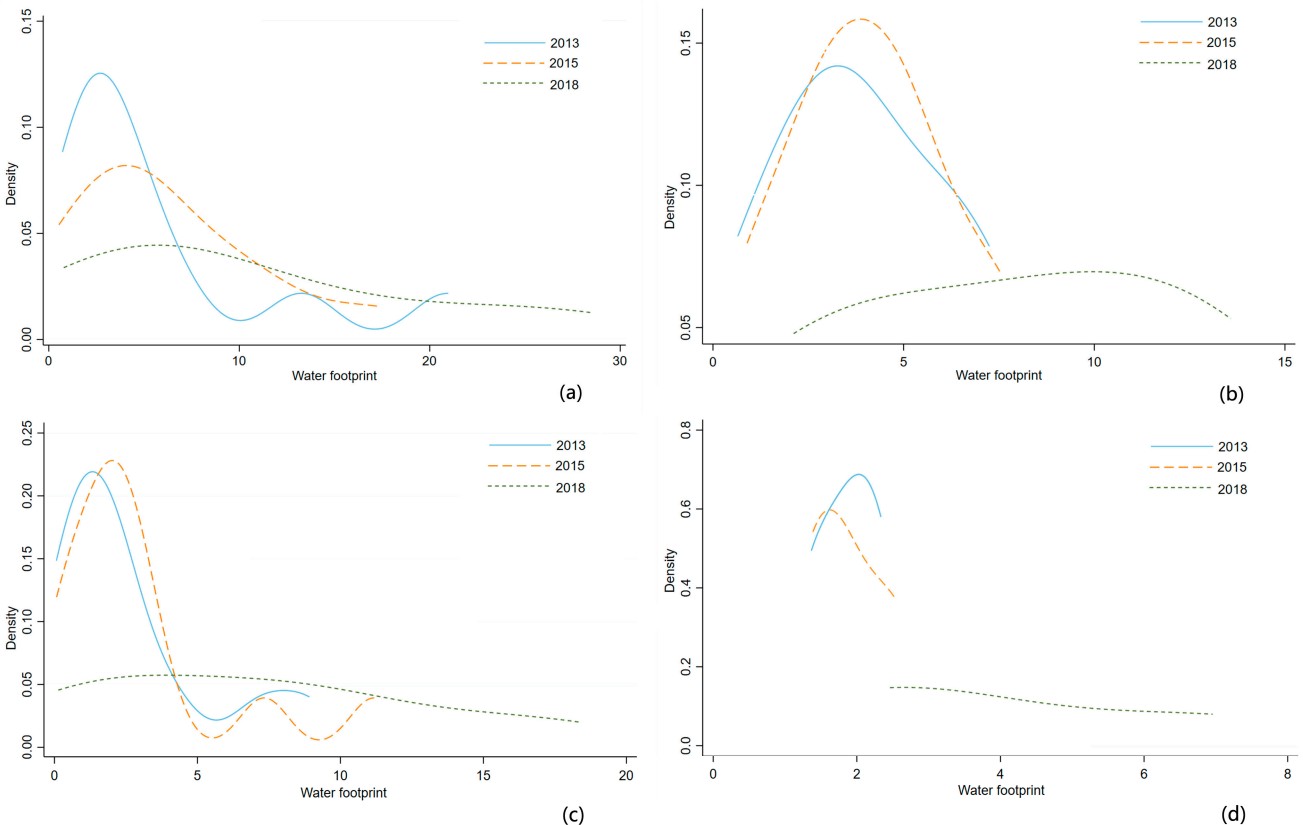

**Figure 7.** (**a**) Dynamic evolution of the kernel density of the *TWF* in the Eastern region; (**b**) Dynamic evolution of the kernel density of the *TWF* in the Central region; (**c**) Dynamic evolution of the kernel density of the *TWF* in the Western region; (**d**) Dynamic evolution of the kernel density of the *TWF* in the Northeast region.

Figure 8 shows that the changes in the shift, peak amplitude, and width of the *TWF* intensity kernel density curve for the Eastern region are all small, indicating that the difference in tourism water use efficiency among the provinces in the Eastern region remained basically unchanged. The overall kernel density curves for the Central and Western regions shift slightly to the left, indicating that the *TWF* intensity slightly decreased and the water use efficiency slightly increased in the two regions. In addition, the curve for the Central region exhibits a bimodal pattern, with the two peaks becoming steeper but the width decreasing and the change interval narrowing, indicating that the polarization pattern of tourism water use efficiency in the Central region gradually deepens, but the absolute difference continuously decreases. The peak increases and the width of the main peak decrease for the Western region, and the difference in tourism water use efficiency among the provinces in the Central region decreases. The kernel density curve for the Northeast region shifts first to the left, then to the right, and eventually back to the right, and the peak increases and then decreases, but there is no significant change overall, indicating that during the study period, the tourism water use efficiency in the Northeast region decreased, but the differences among the provinces remained basically unchanged.

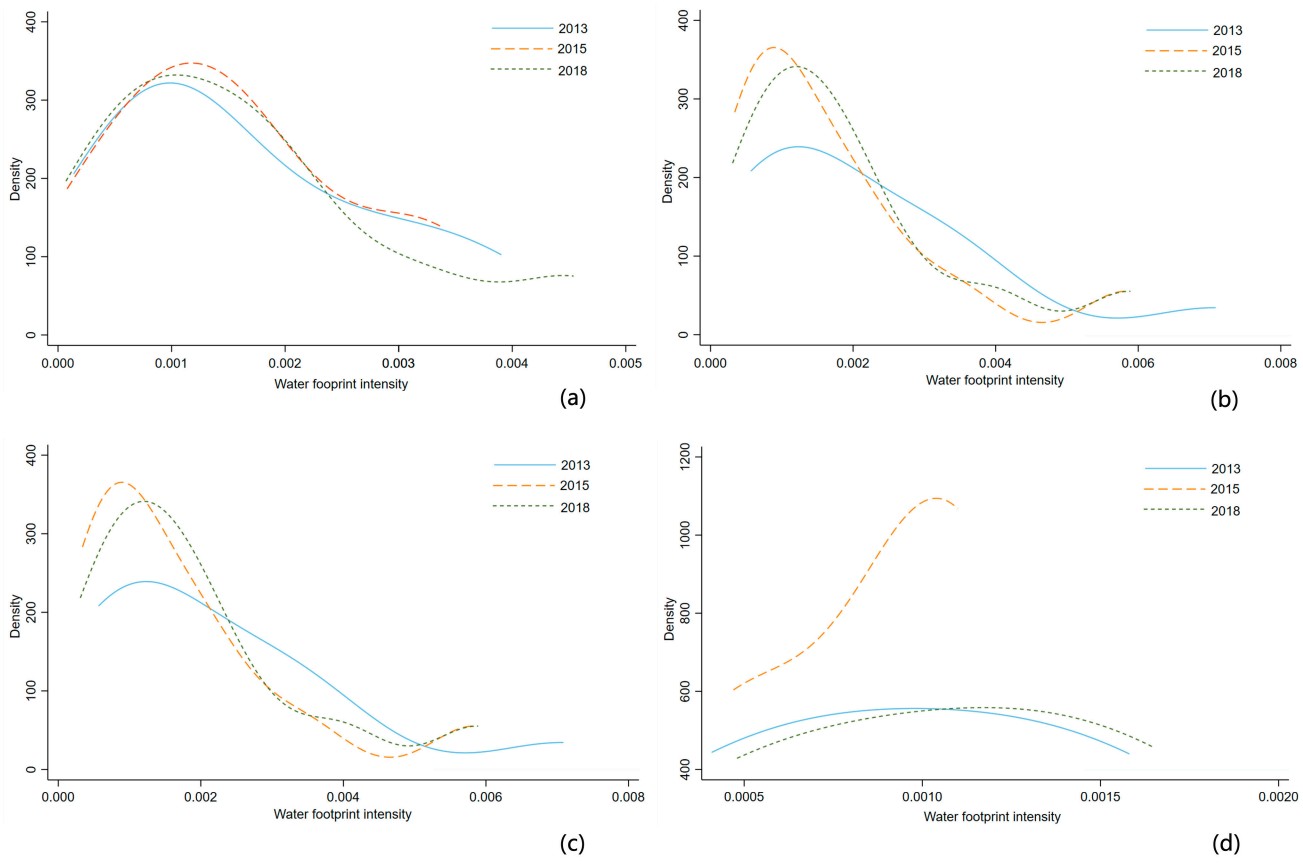

**Figure 8.** (**a**) Dynamic evolution of the kernel density of the *TWF* intensity in the Eastern region; (**b**) Dynamic evolution of the kernel density of the *TWF* intensity in the Central region; (**c**) Dynamic evolution of the kernel density of the *TWF* intensity in the Western region; (**d**) Dynamic evolution of the kernel density of the *TWF* intensity in the Northeast region.

Decomposition of Spatial Differences in the *TWF*

To further investigate the sources of regional differences in the *TWF* of and *TWF* intensity in China, the corresponding Theil indexes for the 2013–2018 period were calculated using Stata software; the results are provided in Table 2.

**Table 2.** Contributions of driving factors of changes in the *TWF* in China in the 2013–2018 period.

| Year | *TWF* | | | | | *TWF* Intensity | | | | |
|---|---|---|---|---|---|---|---|---|---|---|
| | Inter-Regional Disparity | Inter-Regional Contribution | Intra-Regional Disparity | Intra-Regional Contribution | Theil Index | Inter-Regional Disparity | Inter-Regional Contribution | Intra-Regional Disparity | Intra-Regional Contribution | Theil Index |
| 2013 | 0.075 | 17% | 0.360 | 83% | 0.435 | 0.030 | 11% | 0.247 | 89% | 0.277 |
| 2014 | 0.054 | 15% | 0.311 | 85% | 0.365 | 0.032 | 12% | 0.231 | 88% | 0.264 |
| 2015 | 0.071 | 20% | 0.282 | 80% | 0.352 | 0.028 | 10% | 0.247 | 90% | 0.275 |
| 2016 | 0.048 | 15% | 0.269 | 85% | 0.317 | 0.029 | 10% | 0.247 | 90% | 0.276 |
| 2017 | 0.031 | 10% | 0.280 | 90% | 0.311 | 0.031 | 11% | 0.240 | 89% | 0.271 |
| 2018 | 0.032 | 10% | 0.290 | 90% | 0.322 | 0.025 | 9% | 0.255 | 91% | 0.280 |

As seen in the table, the Theil index for China's *TWF* in the 2013–2018 period is significantly positive and shows a gradual downward trend, indicating that the regional differences in China's *TWF* existed and weakened with time. The Theil index for *TWF* intensity remained basically unchanged, indicating that the intensity of regional differences in China's tourism water use efficiency remained unchanged.

In terms of the relationship between inter-regional and intra-regional disparities, both the *TWF* and *TWF* intensity gradually decreased with respect to inter-regional disparities and gradually increased with respect to intra-regional disparities. Therefore, the main

cause of regional differences is the disparity between provinces in the region. We should pay more attention to managing the WF within each region when solving regional WF imbalances.

*3.3. TWF Decomposition Analysis and Recommendations*

From the above analysis, it is clear that the disparity in the *TWF* increased while the disparity in the WF intensity decreased or remained unchanged among the provinces within each region. Therefore, we next focus on the factors that influence the changes in the *TWF* by province. We use the LMDI model to decompose the changes in the *TWF* into three driving factors, namely, WI (WF intensity), PE (per capita expenditure), and TP (number of tourists). In the calculation of the average values, we exclude the influence of abnormal years to reflect the characteristics of various factors in the last two years as much as possible; therefore, the contributions do not add up to 100% for some provinces. The decomposition results for China are shown in Table 3, and the LMDI decomposition results for each province are provided in Table 4.

**Table 3.** Contributions of driving factors of changes in the *TWF* in China in the 2013–2018 period.

| Year | WI | PE | TP |
|---|---|---|---|
| 2013–2014 | −413% | 194% | 320% |
| 2014–2015 | −171% | 19% | 252% |
| 2015–2016 | 9% | 51% | 40% |
| 2016–2017 | 16% | 15% | 69% |
| 2017–2018 | 26% | 13% | 61% |

**Table 4.** Average annual contributions of the driving factors of changes in the *TWF* by province in the 2013–2018 period (%).

| Province | WI | PE | TP | Province | WI | PE | TP | Province | WI | PE | TP |
|---|---|---|---|---|---|---|---|---|---|---|---|
| Jiangsu | 23 | 19 | 58 | Tianjin | 62 | 78 | 74 | Heilongjiang | 19 | 25 | 103 |
| Guangdong | 79 | 6 | −148 | Hubei | 88 | 6 | 26 | Inner Mongolia | 85 | 19 | 21 |
| Zhejiang | 17 | 60 | 63 | Anhui | 69 | 24 | 80 | Jilin | 28 | 33 | 83 |
| Shandong | −6 | 31 | 48 | Shanxi | 21 | 17 | 78 | Hainan | −38 | 47 | 91 |
| Beijing | 79 | 10 | 11 | Yunnan | 43 | 44 | 45 | Gansu | 172 | 9 | 121 |
| Shanghai | 68 | 165 | −134 | Guizhou | 74 | 4 | 29 | Xinjiang | 116 | 29 | 10 |
| Liaoning | −17 | 25 | 92 | Shanxi | 62 | 11 | 27 | Qinghai | 1 | 66 | 114 |
| Henan | 78 | 1 | 21 | Hebei | 7 | 29 | 65 | Xizang | −7 | 61 | 62 |
| Sichuan | −7 | 106 | 72 | Chongqing | 58 | 16 | 26 | Ningxia | −2 | −18 | 120 |
| Fujian | −25 | 38 | 87 | Guangxi | −28 | 29 | 99 | | | | |
| Hunan | 57 | 23 | 20 | Jiangxi | −57 | 50 | 107 | | | | |

3.3.1. Decomposition Analysis of the *TWF* in China

In the 2013–2015 period, the growth of the *TWF* in China was mainly driven by the number of tourists and per capita tourist expenditure; in contrast, the change in the *TWF* intensity reduced the *TWF*. The driving forces, in descending order, were *TWF* intensity, number of tourists, and per capita tourist expenditure. In 2015–2018, the *TWF* intensity, per capita tourist expenditure, and the number of tourists had all positive influences on the *TWF*. A comparison reveals that the degree of the influence of the number of tourists and the per capita tourist expenditure on the *TWF* gradually decreased and that the role of *TWF* intensity changed from inhibition to promotion, with the promotion role increasing year-by-year. The WF intensity reflects the technological level and water use efficiency, and it is a two-way driving force, playing different roles in different time periods. With the development of modernization, the continuous improvement in the technological level has promoted the production of goods to a certain extent, thereby stimulating consumption by tourists and indirectly increasing the *TWF*. On the other hand, the *TWF* intensity decreases, the water use efficiency increases continuously, and the water consumption per unit of

product decreases, which directly reduces the *TWF*. The promotion and the inhibition effects interact with each other and ultimately determine the direction of action of the *TWF* intensity. The promotion role was greater than the inhibition role in the 2013–2015 period, and the inhibition role was greater than the promotion role in the 2015–2018 period. Per capita tourist expenditure reflects the living standards of tourists, and the scale of tourists is the basis of tourism development. With improvements in people's living standards, tourism demand grows, the number of tourists gradually increases, and the consumption of tourism products also gradually increases, which can certainly lead to an increase in the *TWF*. The LMDI decomposition results show that reducing the *TWF* intensity and improving water-saving technologies are key to reducing the *TWF*.

### 3.3.2. Decomposition Analysis of the *TWF* by Province

The level of economic development, amount of resources, and infrastructure conditions vary among provinces in China, and tourist activities differ as well. Such differences inevitably lead to different contributions of different driving factors to the growth of the *TWF*. For example, economic income affects physical consumption by tourists, regional religious beliefs influence the consumption patterns of inbound tourists, and climate and technological levels impact the water production capacity at destinations [82]. As seen in Table 4, the WF intensity of 23 provinces played a promotion role in the *TWF*, and the degree of influence in Gansu and Xinjiang was greater than that in other provinces, indicating that the increase in the *TWF* indirectly generated by technological improvements exceeded the decrease in the *TWF* per unit of product directly generated by technological improvements. Maximizing the direct advantage of technology is key to reducing the WF in these provinces. Except those in Hubei and Ningxia, the per capita tourist expenditures in other provinces all contributed to the *TWF* to some extent. In particular, the degree of influence was higher in Shanghai, Tianjin, and Sichuan than in other provinces; these three are among the developed provinces in terms of tourism and have the highest-ranked annual per capita tourist expenditures, which has a relatively strong promoting effect on the increase in the *TWF* when compared to that in other provinces. The number of tourists positively stimulated the *TWF* in all provincial regions except Xinjiang, Guangdong, and Shanghai. As China's society ages, the proportion of elderly people among inbound tourists has gradually increased. The expenditure structure of elderly people is different from that of young people; therefore, the market vitality gradually declines with the expansion of market size. In addition, the elderly have a strong awareness of water conservation. All these factors lead to a decrease in the *TWF*.

## 4. Discussion

### 4.1. Spatial and Temporal Differences in the TWF

Water is a basic resource for tourism development, as it is needed for cooking, fuel production, garden irrigation, and park construction, among other needs. Both water pollution and water shortages can damage the image of a tourism destination [83]. Herein, the changes in China's *TWF* and the influencing factors are analyzed in-depth at three scales, i.e., national, four regions, and 31 provinces. The results of this study essentially explain the difficulty in balancing tourism development and tourism WF reduction and provide a reference for adopting targeted regional water management strategies and coordinating inter-regional water resource allocation. In addition, the present study also offers a new perspective on promoting high-quality regional tourism development from the lens of the WF.

The results indicate that the *TWF*s in China and the four regions increased year-by-year. Although not all the end consumption items are covered in the calculation process, and there are differences in the calculation methods, the total values are still much higher than those for other countries, such as Spain, Japan, or Australia [13,54]. The *TWF* in 2017 was $199.42 \times 10^8$ m$^3$, which is similar to the result ($191.95 \times 10^8$ m$^3$) of Lee et al. [22]. The increase in the Western and Northeast regions was higher than that in other regions

due to the impact of policies such as China Western Development and the Revitalization of the Old Industrial Bases of Northeast China. We also found that the growth rate of the *TWF* was consistently larger than that of tourism revenue and that the two were not synchronous, indicating that tourism water consumption was not effectively controlled in the 2013–2018 period. This unsynchronized trend also appeared in the development of West Lake tourism; tourism revenue increased by 21% while tourism waste increased by 55% [18]. This series of phenomena is contrary to the goal of water conservation in tourism. Tourism water use efficiency improved in all regions except the Northeast region, a result we speculate was due to the yearly decline in GDP growth rate in the Northeast region after 2013, leading to a decrease in water conservation-related inputs. Deng [84] calculated the blue, green, and gray water footprints of various industries, and the *TWF* value is much lower than that of agriculture, construction, and food and beverage industries. However, the ever-increasing number of tourists will still bring indispensable pressure on water resources to water-scarce tourist destinations.

In terms of spatial distribution, the *TWF* in provinces to the southeast of the Hu line was higher than that in provinces to the northwest of the Hu line. The northwest inland areas are farther from the sea, with an arid climate and low precipitation, thereby limiting water consumption to a certain extent due to the insufficient supply of water resources. The southeast provinces, in addition to their geographical location, have a higher level of urbanization, high transportation accessibility, and more developed tourism, thereby resulting in a high tourism water demand. However, the *TWF* intensity in southeast provinces was significantly lower than that in northwest provinces, which we speculate is due to the influence of regional economic development. Therefore, it is particularly important to reduce the water consumption of the urban agglomeration to the southeast of the Hu line and to focus on improving water use efficiency in the Northwest region while ensuring economic development.

The kernel density estimation and Theil index results indicate that during the observation period, the *TWF*s of the four regions increased, and the intra-regional absolute difference among provinces gradually increased; however, the inter-regional difference gradually decreased. In recent years, tourism has been developing rapidly in some provinces in each region, such as Hunan and Hubei in the Central region, Chongqing, Sichuan, and Shaanxi in the Western region, and Liaoning in the Northeast region. Although the changes in the *TWF* intensity in the four regions were different, both inter-regional and intra-regional differences remained basically unchanged or slightly decreased. In addition, the main cause of the inter-regional differences was the changes in the differences between the provinces within each region. There is currently no research on the regional differences in tourism water footprints of various provinces in China. However, Wu et al. [85] found that the spatial difference in China's WF is mainly due to the regional differences in consumption, while the contribution of intra-regional differences is relatively low. Therefore, reducing the inter-regional difference in the *TWF* of each province is the key to achieving spatial equilibrium.

### 4.2. TWF Driving Factors

LMDI analysis allowed the identification of the driving forces that affected the changes in the *TWF* of each province, based on which tourism water use policies can be formulated to reduce regional disparities. Sun and Liu [18] found that the number of tourists and tourism income in West Lake increased water pollution, which means that the grey water footprint increased. Similarly, the growth of the *TWF* in China was mainly driven positively by the number of tourists and per capita tourist expenditure; however, the driving effect weakened continuously, probably due to the increase in the proportion of elderly tourists and the change in the structure of tourist expenditure. The role of *TWF* intensity shifted from inhibition to promotion, which means that the technical elements have produced phased effects. However, Lee et al. [29] show that technical elements have always had a promoting effect. Through comparison, it is found that *TWF* intensity and direct con-

sumption coefficient can represent technical effects, but different indicators will produce different results. Therefore, the current priority for China should be to further increase investment in technology to improve water use efficiency and achieve economies of scale in water use.

Using the average annual *TWF* and average annual *TWF* growth rate for each province from 2013 to 2018 as characteristic variables, the 31 provinces in China were divided into five clusters (Figure 9). Cluster 1 was characterized by a large value and a high *TWF* growth rate. Cluster 2 was characterized by a small value but a high *TWF* growth rate. Cluster 3 was featured by a large value but a low *TWF* growth rate. Cluster 4 was characterized by small value and low *TWF* growth rate. Next, we discuss the characteristics of each cluster and propose suggestions for reducing the *TWF*.

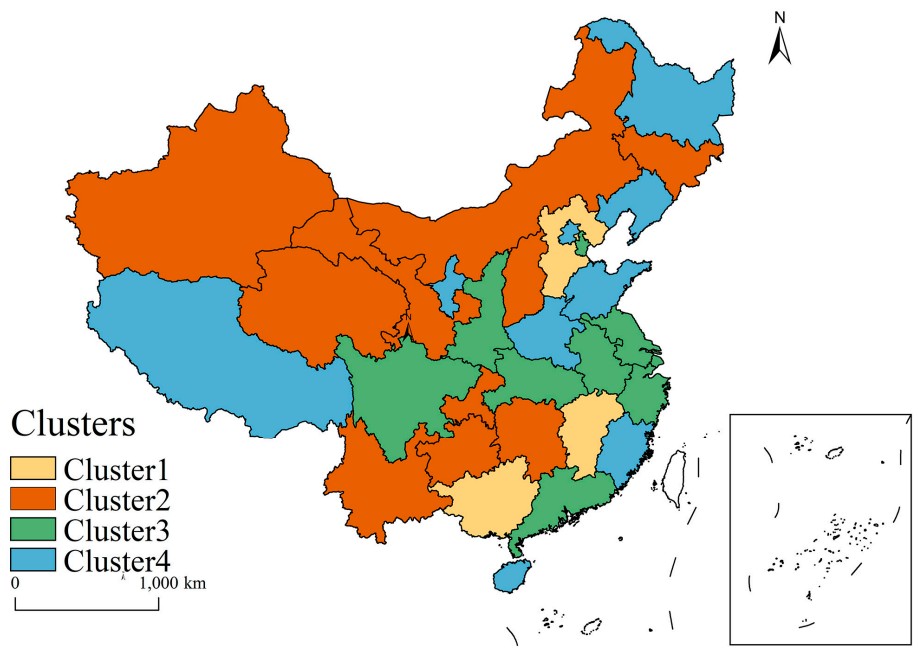

**Figure 9.** Classification of the 31 provinces in China.

Cluster 1 consists of Hebei, Guangxi, and Jiangxi, with *TWF* values slightly higher than the national average but with growth rates significantly higher than the average. Unlike the changes in *TWF* intensity, per capita expenditure, and the number of tourists, which had positive effects in Hebei, the decrease in *TWF* intensity had an inhibiting effect in Guangxi and Jiangxi. Among the three driving factors, the expansion of the tourism scale played a dominant role in the growth of the *TWF* of the three provinces. In addition, the members of the cluster ranked in the middle in terms of tourism revenue and the number of tourists, the growth rates of which, however, both ranked high. These provinces require special attention. We believe that the *TWF*s of the three provinces will continue to grow rapidly with the expansion of the tourism market. The industrial structure can be adjusted for tourism development in Hebei, a heavy industrial province, and tourism is a favorable means to promote economic development in Guangxi and Jiangxi, two underdeveloped provinces. Overall, this cluster should vigorously develop tourism while focusing on the promotion of tourists' water conservation awareness, financial subsidies for water-saving enterprises, and increased investment in research and development of water-saving facilities.

Cluster 2 is composed of Hunan, Shanxi, Jilin, Yunnan, Guizhou, Chongqing, Inner Mongolia, Gansu, Xinjiang, Qinghai, most of which are located in southwest and northwest China and were characterized by a small value but high *TWF* growth rate. The *TWF* growth rate was higher than the national average. In particular, Guizhou had a *TWF* growth rate that was 2.25 times the national average but had a WF that was significantly lower than

the national average. In addition, changes in *TWF* intensity, per capita expenditure, and the number of tourists all contributed positively to *TWF* growth. China has introduced a series of policies to promote the economic development of its Western region, but factors such as underdeveloped transportation and poor infrastructure construction have been restricting the development of various industries, including tourism, and the size of the tourism market is still small compared with that of developed coastal areas. However, since the implementation of China Western Development, tourism construction funds have been continuously funneled to western China, and transportation networks (e.g., railways and highways) and infrastructure (e.g., power generation and communication) in western China have undergone continuous improvements, attracting an increasing number of tourists to western China as a tourism destination. We estimate that the dividends of China Western Development will continue to emerge, which means that the *TWF* of these provinces will continue to increase rapidly for a long time to come. Considering the important role of tourism in the economic development of western China, the water resource management strategy proposed for this cluster is to establish a water conservation-oriented tourism development model and to set strict water conservation goals for various sectors.

Cluster 3 consists of Tianjin, Jiangsu, Shaanxi, Hubei, Zhejiang, Anhui, Guangdong, Shanghai, and Sichuan, with a large value but a low WTF growth rate. From 2013 to 2018, each province had an average annual WF higher than the national average but a growth rate lower than the average. In particular, the WFs of Guangdong and Shanghai were $20.61 \times 10^8$ m$^3$ and $14.45 \times 108$ m$^3$, respectively, far above the average. The total *TWF* of this cluster accounted for 52.5% of the national total, making it a key area for water conservation and emissions reductions. For Shanghai and Guangdong, changes in per capita tourist expenditure and *TWF* intensity increased the *TWF*; however, the increase in the number of tourists had the opposite effect, which we speculate might be caused by changes in the structure of tourists. Based on relevant data, as of 2018, China's population aged 60 years and older accounted for 17.9% of the total population, elderly tourists accounted for more than 20% of the total number of tourists in the country, and the size of the tourism market grew year-by-year. Guangdong and Shanghai, as tourism hotspots, were inevitably the first to be affected by the increase in the proportion of elderly tourists. For Sichuan, the inhibiting effect of WF intensity was stronger than the promoting effect. For other provinces, changes in the three driving factors all had a positive effect on the *TWF*. These provinces have relatively mature tourism markets and can take the lead in setting water use caps for tourism-related sectors, attracting elderly tourists while ensuring a certain level of tourism market vitality, guiding tourists to consume "water-saving products," and thus promoting the restructuring of the industry to sectors with a low *TWF* intensity.

Cluster 4 is composed of eight provinces with small *TWF*s ($0.70 \times 10^8$ m$^3$ to $4.32 \times 10^8$ m$^3$) and low WF growth rates (3.81% to 15.65%), both of which were lower than the respective national averages, indicating that tourism development generated little pressure on the local tourism water environment. The changes in the per capita expenditure and the number of tourists of all provinces in this cluster, except Ningxia, increased the *TWF*. The WF intensity had a promoting effect in Heilongjiang, Beijing, and Henan but had an inhibiting effect in other provinces. These locations can further encourage tourism development and increase tourism revenue; however, it is necessary to increase investments in technology, improve the efficiency of water resource use, and fully consider the inhibiting role of technological effects.

## 5. Conclusions

Tourism activities break the regional boundaries of water demand and affect the regional water supply and demand balance. Characterizing the pattern of water resource pressure caused by tourism development on a national scale is conducive to the macro-control of sustainable tourism development. In the present study, China, the second-largest economy in the world, is used as the study area to investigate the spatial and temporal

patterns of water consumption and the driving factors behind tourism development from the WF perspective and to make targeted recommendations for optimization. The total *TWF* of mainland China doubled during the 2013–2018 period, and the *TWF* decreased at the national scale but changed (increased or decreased) inconsistently at the provincial scale, with significant differences. The WF of accommodation and food has always accounted for the bulk of the *TWF* in mainland China, and its proportion continues to increase; however, inter-provincial differences in the structure of the *TWF* still exist.

The LMDI model was used to attribute the changes in China's *TWF* to three factors, namely, WF intensity, per capita expenditure, and the number of tourists. In particular, the role of WF intensity is constantly prominent and has become an important factor in the increase in the *TWF* in China. Therefore, the implementation of water conservation policies and technological innovation are important ways to control the total *TWF*.

In terms of the calculation method, different from previous studies, this research converts the currency flow of various departments into the material flow so as to provide a more detailed data basis for tourism water policy formulation and tourism ecological benefit analysis. In addition, most regions in the world do not have tourism satellite accounts (TSA), and the lack of tourism statistics has hindered the output of *TWF* research to a certain extent. The IO analysis in this paper can effectively solve this problem. In terms of research perspectives, this article breaks through the limitations of static research or large-scale research and deeply analyzes the characteristics of spatial differences and temporal changes of *TWF*. IO analysis facilitated the investigation of the overall *TWF*, but the IO model also has limitations [61,86]. The IO tables for each province in mainland China are published every five years, but the IO tables for some provinces in 2017 were unavailable. Considering the relative stability of IO coefficients, the present study used the IO tables of 2012 to estimate the *TWF*s for the 2013–2018 period. There are no data on the structure of domestic tourism consumption in provincial statistical yearbooks, and thus, the inbound tourism consumption structure is used instead, which fails to distinguish the WFs of domestic and inbound tourists. In addition, due to data limitations, the present study did not establish a link between tourist characteristics and WF, preventing a micro-level investigation. Therefore, in future research, the WF generated by different types of tourists from different places of origin should be investigated in more detail using the research framework of the present study.

**Author Contributions:** Conceptualization, P.M.; methodology, Z.X.; software, Z.X.; validation, P.M.; formal analysis, Z.X. and W.H.; data curation, C.A., L.Y. and F.Z.; writing—original draft preparation, Z.X. and W.H.; writing—review and editing, P.M.; visualization, C.A., L.Y. and F.Z.; supervision, P.M.; project administration, P.M.; funding acquisition, P.M. All authors have read and agreed to the published version of the manuscript.

**Funding:** This research was funded by the Key Laboratory Projection of Sustainable Development of Xinjiang's Historical and Cultural Tourism, Xinjiang University, grant number LY2020-06; it was also funded by the National Natural Science Foundation of China, grant numbers 41961038, 41661106, and 41461111.

**Institutional Review Board Statement:** Not applicable.

**Informed Consent Statement:** Not applicable.

**Data Availability Statement:** Tourism related statistics can be collected from the China Economic and Social Big Data Research Platform (https://data.cnki.net/yearbook (accessed on 16 September 2021)). The data presented in this study are available on request from the corresponding author.

**Acknowledgments:** We acknowledge all the data source providers of this paper. We are also grateful to the editor and the reviewers for their helpful comments.

**Conflicts of Interest:** The authors declare no conflict of interest.

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
