# Peer review of "Spatial and Temporal Differentiation of the Tourism Water Footprint in Mainland China and Its Influencing Factors"

_sustainability, doi:10.3390/su131810396_

Round 1

Reviewer 1 Report

The ms

"Spatial and temporal differentiation of the tourism water footprint in mainland China and its influencing factors"

by Xiao et al.

shows the trends of change in the water footprint of tourism in China in the years 2013-2018 using the input-output analysis. In particular, it analyzes regional differences in changes in tourism water footprint using the kernel density estimate and Theil index. Finally, it investigates the driving factors of the spatial and temporal differentiation of the tourism water footprint using the logarithmic mean of the Divisia index model.

Therefore in my opinion it fits the goals of Sustainability journal, it may be of interest to a vast community of users and deserves being published in this journal.

Overall, this is a clear, concise and well-written manuscript.

The introduction is pertinent and based on interesting papers.

The procedure is described in details and gives sufficient information on the study logic.

The methods are generally very helpful and are appropriate.

In addition, the results are clear.

Moderate English changes required. Please carefully review the figures in the manuscript.

L 34: please replace "accelerate" with "will accelerate";

L 36: please replace "if considering other factors" with "if other factors are considered";

L 108 & L 110: please replace "product production" with a different wording; 

L 258: please replace "is" with "is define";

Figure 1 (a & b): please add the labels to the axes with the relative units of measurement;

Figure 2: please add the labels to the axes with the relative units of measurement; the percentages are not legible, please change the color (e.g. black?);

On page 10 figure 3 is repeated twice: at the top it includes only part (b), at the bottom part (a) and (b);

Figure 3 (a & b): At the top right there is a box, the purpose is not clear. Please add a scale bar and units of measurement in legend (Figure b);

Figure 4 (a & b): please add the labels to the axes with the relative units of measurement;

L 480: please replace "number" with "the number";

L 480: please replace "all had" with "had all";

L 487: please replace "product production" with a different wording;

L 575: please replace "is key to achieving" with "is the key to achieving";

Author Response

Dear reviewer,

Thank you for your comments, which are all valuable and very helpful for revising and improving our paper. Please see the attachment.

Sincerely,

Polat Muhtar

Reviewer 2 Report

The topic has an outstanding significance but the article has some serious shortcomings:

  • spellcheck and the check of references are both needed;
  • the authors should take into account that many readers are not familiar with certain abbreviations, concepts (e.g. WF, TWF, blue, green, and grey WFs, the meaning of China Regional Input-Output tables, Hu line, etc.) These concepts should be presented and defined in the text.
  • The four regions should be presented on a map (line 335).
  • Figure 3. is placed oddly in the text: on the top page 10 and the bottom of this very same page as well.
  • The authors should provide more information on Figure 3 or in the title of this figure. For example, the time of the data should be indicated on the figure. Furthermore, there is an element on both the a and be part of Figure 3, at the last map that is missing from the previous two maps.
  • the logic of data and methods part should be changed: first the data should be presented, then the methods should follow.
  • The Discussion is not a real discussion; it presents the results but does not compare them to the previous researches. Thus the results are only described but they are not discussed.
  • The authors should highlight their contribution to methodology, concepts, theory etc. What is the general contribution to scholarship?
  • The method of clustering mentioned on page 16 is not clear: is it a result of a cluster analysis? Then the method should be described and the clusters should be described in a more detailed manner.

Author Response

(The authors gave the same response as above.)

Reviewer 3 Report

Dear and respective authors,

The manuscript entitled Spatial and temporal differentiation of the tourism water footprint in mainland China and its influencing factors is a valuable study which provides interesting results related to the contemporary issues that encompass both, ecological safety aspects (water scarcity - water footprint) and environmental pressures induced by the tourism industry. Thus, spatial and temporal differentiation of the characteristics of water consumption at tourism destinations for China’s mainland was addressed in this research by the respective authors. Although the presented work with its valuable methodology and results deserve to be considered for publishing in the journal Sustainability, it still has some issues needed to be addressed before this step. Below is the list with my suggestions for manuscript enhancement.
1. In the section 1. Introduction, on page 1, line 44, please insert adequate reference at the end of the sentence: “It is estimated that by 2030, the Earth will face a 40% water shortage”. Page 2, line 47, separate “[9].” from the beginning of the sentence “Tourism…”.

  1. For the introduction part, I advise authors to see the paper of Sun, Q., Liu, Z. 2020. Impact of tourism activities on water pollution in the West Lake Basin (Hangzhou, China). Open Geosciences 12, 1302–1308. https://doi.org/10.1515/geo-2020-0119 because there are some nice sentences and aspects related to this research that can be implemented into this introducionary section and in 4. Discussion part as well.
  2. In the section 2. Materials and Methods, it is highly advised to the authors to include an additional figure - work flowchart related to data and methodology after sub-chapter 2.2.3. It can certainly help the wider audience to visualize the research team effort of the database products derived from the complex number of step procedures used in this research. Also, the authors need to mention which software was used for graphical and spatial representation of the derived results, mainly thematic maps (e.g. ArcGIS etc).
  3. In the subchapter 3.2.1.1. Total difference, page 9, line 366m please insert the missing text into bracket. Also, in the line 368 “sity in China from 2013 to 2018 showed the following three characteristics.” please correct the typo “sity” into “city”. Figure 3. is missing its right part of the map, please revise this technical issue. Is this figure has more features since only (b) part is presented here??? Please check and revise.
  4. On page 15, line 527, please un-italic the chapter 4. Discussion, bold it and make it a main chapter part of the manuscript.
  5. I suggest to authors to present “Table 5. Classification of the 31 provinces in China” as a thematic map, since it could be more convenient for the readers to better illustrate the given clusters.
  6. Concluding remarks are fine.
  7. Reference list is good, but can be slightly enhanced (please see comment no.2).
  8. I suggest the English proofread of the manuscript and thorough check up on the typos within the text.
  9. On page 19, lines 690-696, please use only author initials as proposed by the MDPI guidelines.

    After the above mentioned suggestions for improvement are implemented into the manuscript, I highly suggest to Editorial Board of the Sustainability to consider it for publication. It certainly has great potential considering the research field and used approaches.

    Kind regards!

Author Response

(The authors gave the same response as above.)

Round 2

Reviewer 2 Report

The article is improved compared to the previous version, most of the issues raised in my previous review are settled. Only a few remaining issues can be highlighted:

  • the most important issue is that the Conclusion should highlight the overall contribution to scholarship in a more explicit and direct way
  • I think the nationality of a scholar is not significant (see line 101)
  • the problems mentioned in lines 62-63 are not limited to world heritage sites, these are general problems, affecting various tipes of sites
  • I do not feel that the flowchart should be in a separate subchapter (2.2.6.)

Author Response

Dear Rviewer

Thanks for your kindly help in our previous manuscript.
